# The Outdoor Field Test and Energy Yield Model of the Four-Terminal on Si Tandem PV Module

**Kenji Araki** [1,*] **, Hiroki Tawa** [2] **, Hiromu Saiki** [2] **, Yasuyuki Ota** [3] **, Kensuke Nishioka** [2] **and Masafumi Yamaguchi** [1]

[1] Semiconductor Laboratory, Toyota Technological Institute, Nagoya 468-8511, Japan; masafumi@toyota-ti.ac.jp
[2] Faculty of Engineering, University of Miyazaki, Miyazaki 889-2192, Japan; hk14028@student.miyazaki-u.ac.jp (H.T.); hk14017@student.miyazaki-u.ac.jp (H.S.); nishioka@cc.miyazaki-u.ac.jp (K.N.)
[3] Institute for Tenure Track Promotion, University of Miyazaki, Miyazaki 889-2192, Japan; y-ota@cc.miyazaki-u.ac.jp
[*] Correspondence: cpvkenjiaraki@toyota-ti.ac.jp; Tel.: +81-52-8091830



**Featured Application: This technology is expected to be applied to high-performance photovoltaic applications like zero-emission buildings, light-weight aerospace, and possibly, vehicle-integrated photovoltaic.**

**Abstract:** The outdoor field test of the 4-terminal on Si tandem photovoltaic module (specifically, InGaP/GaAs on Si) was investigated and a performance model, considering spectrum change affected by fluctuation of atmospheric parameters, was developed and validated. The 4-terminal on Si tandem photovoltaic module had about 40% advantage in seasonal performance loss compared with standard InGaP/GaAs/InGaAs 2-terminal tandem photovoltaic module. This advantage increases (subarctic zone < temperate zone < subtropical zone). The developed and validated model used an all-climate spectrum model and considered fluctuation of atmospheric parameters. It can be applied every type of on-Si tandem solar cells.

**Keywords:** photovoltaic; solar spectrum; tandem cell; energy yield model; on-Si tandem; terminal

## 1. Introduction

High efficiency is the typical research target of photovoltaic technology. However, it is also known through field experience and theoretical analysis considering spectrum fluctuation that the photovoltaic system that wins the race of efficiency does not always gain the best yield in the real installation [1–4].

Currently, the Si solar cell has commonly prevailed in the market. The best efficiency of the Si that was confirmed by testing laboratories is 26.7% [5], and the theoretical limit is 29.43% [6]. For further improvement of efficiency, multi-junction or tandem configuration is preferred. The idea of multi-junction cells was suggested [7] and investigated [8] in the early days of photovoltaic technology. Significant progress was triggered by AlGaAs/GaAs tandem cells with tunnel junctions [9] and metal interconnections [10–12]. At that moment, it was predicted that the power-conversion efficiency of multi-junction solar cells would reach close to 30% [13], but this was not fulfilled due to difficulties in stable tunnel junctions [14] and defects in the AlGaAs [15]. The break-through were high-performance stable tunnel junctions with a double-heterostructure [16]. InGaP was introduced for the top cell [17], and as a result, it finally achieved almost 30% efficiency by a GaInP/GaAs cell [18]. Higher efficiencies have been made with InGaP/GaAs/InGaAs triple-junction cells [19] and with a 5-junction cell [20].

On-Si tandem solar cells use the widely-used Si solar cell as the bottom junction of the tandem solar cell. Because the technology of Si solar cells is well-established, the production cost of the solar cell is expected to reduce; at least, the one to the substrate or the bottom junction [21]. The III-V/Si (III-V on Si) 3-junction and 2-junction tandem solar cells right now exhibit excellent efficiency with 35.9% [22] and 32.8% [22]. The perovskite on Si 2-junction tandem solar cells reached 28.0% [23]. The efficiency of CdZnTe on Si tandem solar cells was 16.8% [24], and that of GaAs nano-wire on Si tandem solar cells was 11.4% [25]. The III-V/Si tandem solar cell has the best efficiency among them. Related to this III-V/Si tandem solar cell technology, there are several module technologies, including a partial concentrator module [26]. The tested power conversion efficiency of a pair of InGaP/GaAs partial concentrator cell on Si, including optical loss of the concentrator optics, was 27.1% [27]. The partial concentrator module, also using InGaP/GaAs partial concentrator cell on Si and designed for automobile applications, showed 21.5% module efficiency [28]. The non-concentrating module using InGaP/GaAs cell on InGaAs by 4T configuration showed 31.17% [1,29].

Another big group of the candidates of the on Si tandem cells is Perovskite on Si tandem cells, including a review of the 4T Perovskite on Si cell [30] and challenging the growth on the textured silicon [31–33]. Related to the topic of this paper, consideration of the performance of the Perovskite on Si tandem cells was done, extending the standard testing condition (AM1.5G spectrum and consideration of diffused sunlight). Until now, it had not been validated by annual outdoor operation [34].

Regardless of the material type combined with Si bottom junction, mismatching loss by the spectrum change is a big issue [35–39]. The intensive studies of the spectral sensitivity of the tandem cell were firstly done in the analysis of the concentrator photovoltaic (CPV), since tandem solar cells were expensive. Nevertheless, the high performance and the concentration operation were commercially advantageous to the application [40–49]. The spectrum shift from the standard AM1.5G spectrum destroys the balance of the output electrical current from the sub-cell. Then, the sub-cell with the least output current constrains the total electrical current by the conservation of the carrier or the Kirchhoff's law. It is called "spectrum-mismatching loss". It is an inherent loss for every type of tandem solar cell, regardless of CPV or standard flat-plate application, irrespective of on-Si or on-CIGS tandem, and regardless of III-V or Perovskite tandem solar cells, except for more than three terminal configurations where the output of the sub-cells is individually connected to the load. It is essential to note that a variation of the solar spectrum by sun height and fluctuation of scattering and absorption of the air by the seasonal effect will affect the annual performance. However, a lot of research and development was done to minimize the influence of spectrum change by the improvement of the solar cell design [50–55].

Research and development seeking for the device design with the robustness to spectrum change have been made in the past 20 years, including a computer model named Syracuse by Imperial College London [2,56,57]. For CPV applications that have been the center of the study of the tandem cells with the robustness of spectrum issues, it was understood that the chromatic aberration of the concentrator optics enhanced the spectrum-mismatching loss [58–62]. However, such loss could be solved by the innovation of optics, including homogenizers and the secondary optical element (SOE) [63,64]. The remaining problems of the spectrum-mismatching loss have been overcome by the adjustment of the spectral response of each sub-cell, including overlapping the absorption spectrum and broadening the absorption band to the zone of massive fluctuation.

The model, theory, and advanced design of CPV, as well as innovations for solving the issue of spectrum loss in the real-world solar irradiance, were tried to be validated by outdoor energy yield monitoring. However, it was not very successful. One possible reason was that the performance of CPV was also sensitively affected by the tracking error and misalignment of the concentrator optics and the solar cell. However, this issue was solved by the series of innovations of the testing method of the misalignment [65], validation in the production line of the CPV module [66], and the analysis of the impact on the energy output [67,68]. After solving uncertain factors, the influence of the spectrum change was re-examined. However, it was found that the standard model using air-mass (or the sun height) as the parameter of the solar spectrum, which has been used by many scientists and for more

than 20 years, could not explain the seasonal trend of the performance by analysis [69] and the same results using variance (ANOVA) [70]. The new findings that directly and more strongly influence the spectrum than the air-mass was a fluctuation of atmospheric parameters. This was because the impact by the sun height appears strongly when the airmass value is more than 2, and both summer and winter have the early morning and the late evening time equally. Therefore, without consideration of atmospheric parameters, it is almost impossible to predict the annual energy yield; in other words, it is almost impossible to design the optimal tandem solar cells in yearly energy yield.

Recently, tandem solar cells have been considered for use in non-concentrating applications, out of CPV, including vehicle-integrated photovoltaic (VIPV) and car-roof photovoltaics (PV) [71–82]. It was thought that most electric vehicles (EV) might be able to run by solar energy using tandem cells on the car roof [1]. The area of the car roof and other car-bodies are limited. Moreover, solar cells cannot be laminated to an undevelopable curved surface of the car body. It is difficult to cover the restricted area of the car-roof surface entirely. Therefore, extremely high performance that can be brought by tandem solar cells is highly required.

The analysis of the spectrum sensitivity on CPV was done in our previous research [83,84]. The calculation and analysis for CPV were more straightforward because we did not have to consider angular effects combined with the mixture of the direct and diffused components of the sunlight. Different from CPV applications, in which the cell is always perpendicularly illuminated by the sun using a solar tracker and only utilizes the direct sunlight, the typical application needs to use a diffused component of the sunlight from the sky, ground reflection, and skewed solar rays, with a combination of direct and diffused elements as a function of the sun orientation relative to the solar panel orientation. For an extension to non-concentrating applications, we need to solve the complicated coupling of spectrum and angles (Table 1). The key parameters are atmospheric parameters that are dependent on each other. For example, different incident angle modifier and different orientation lead to a diverse mixture of direct and diffused sunlight.

**Table 1.** The difference in performance modeling between concentrator photovoltaic (CPV) and standard installation [1]. The outdoor performance of CPV was intensively studied in more than 20 years considering the spectrum mismatching problem of the tandem solar cells.

| | CPV [1] | Normal Installation |
|---|---|---|
| Solar spectrum | Only direct | A mixture of direct, diffused from the sky, and reflection |
| Angle | Always normal | Varies by time and seasons |
| Spectrum by angle | Constant (only normal) | Needs consider coupling to angle |

[1] It only generates power by direct solar irradiance using a 2-axis solar tracker.

Four-terminal (4T) tandem cells were designed so that the output of the Si cell and other top junctions were taken independently, thus robust to the spectrum change. In the case of III-V/Si three-junction solar cells that are frequently considered as an excellent candidate for high-efficiency solar cells, the output terminal from the top cell comes from two-junction III-V solar cells, and the top two-junction cell is still susceptible to spectrum change. Therefore, a long-term field test of the module using III-V/Si 4T solar cells is essential to the validation of the use of this configuration. Currently, the best efficiency confirmed by the third-party is 33.3% [85].

Other groups of the 4T type tandem cells are partial concentrator cells and 3T tandem cells. The partial concentrator solar cell was first proposed in 2017 [86], using wide-acceptance concentrator optics that selectively concentrate onto the top III-V cells stacked on the larger size Si bottom cell. This type of tandem solar cell was considered for the application to VIPV [87] and various design methods and models were invented [88–90]. Several prototypes were made [28], and an annual average efficiency as high as 27.4% was anticipated [77]. 3T type tandem cells basically connect one of the terminals of the pairs of the two-terminal. This method has an advantage of simplifying the interconnection of the solar cells among the PV module. There are several types of tandem cells, including Ge-based III-V

cell [91], polymer type [92], III-V monolithic [93], and III-V on Si [94]. The advantage of the better spectrum matching operation using this 3T configuration was also studied [95].

The background of this study and our motivation are summarized as follows:

1.  Tandem solar cells are highly efficient, and various types of the device structure were studied. On-Si tandem is one of them, and it has a distinct advantage of cost, using well-established Si solar cell technology.
2.  Regardless of the type of material, the annual performance of the tandem solar cells does not perform well due to spectrum mismatching loss.
3.  The modeling of the spectrum mismatching loss was studied relying on the airmass variation. The intensive study on CPV performance in more than 20 years revealed that the fluctuation of atmospheric parameters played an essential role.
4.  Due to the development of the new application of the high-efficiency solar cell, including vehicle-integrated solar cells, the precise annual energy yield modeling of the tandem solar cells is required. The knowledge on precise spectrum-mismatching modeling in CPV is expanded to the non-concentration standard installation.
5.  4T on Si tandem solar cell is a good candidate for the robustness to the spectrum variation. Its outdoor operation and energy yield modeling was intensively studied in this article. The model did not rely only on airmass but considered real fluctuation of the spectrum in all kinds of climate, considering atmospheric fluctuation.

## 2. Methods

The purpose of our work is to develop an accurate way of predicting the performance of photovoltaic modules using tandem 4T solar cells, considering spectrum variation. The base model of the spectrum is Bird's model [96]. Bird's model only covers clear sky days, and we expanded it to roughly cover all climates [97]. This new spectrum model and the response of the tandem module were validated by the long-term (multiple years) measurement of the performance of the module [97].

### 2.1. Device Configuration

The fundamental difference of 2T and 4T configuration is that the former connects all sub-cells in a series, and the latter divides the circuit into two pieces using two pairs of output lead, namely, the four-terminal (Figure 1). Each pair of leads can be connected to two independent loads so that each 2T output can be controlled by independent MPPT (maximum power point tracking). There are two advantages. One is less loss in spectrum mismatching because the shorter-wavelength zone and longer-wavelength zone are connected independent loads. Another is more flexibility in the design of the solar cell. In this particular case, the 4T configuration uses Si in the bottom sub-cell and a III-V two-junction cell in the top sub-cell. Both sub-cell technologies are well-established, and it is not necessary to consider various constraints of the integration into one piece (typically monolithic growth) solar cells.

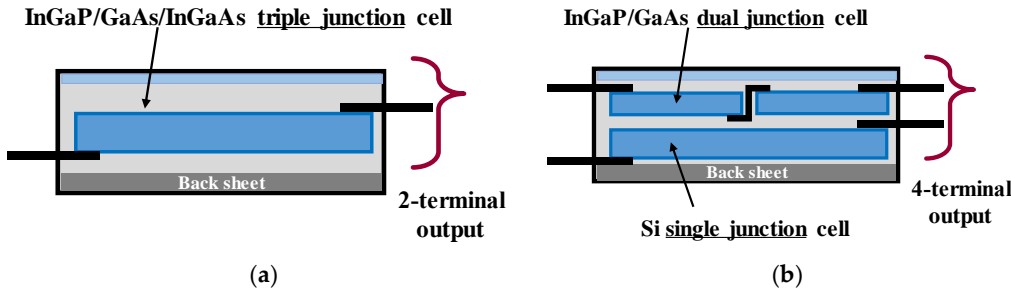

**Figure 1.** Measured 4-terminal (4T) III-V/Si module and its solar cell structure: (**a**) Description of 2-terminal (2T) III-V 3-junction module; (**b**) Description of measured 4T III-V/Si module, 2 + 1 junctions. Note that the number of junctions is the same.

### 2.2. Measurement System

The field test was done at the University of Miyazaki. The details of the measurement setup can be found in our recent publication of Applied Sciences [97], as shown in Figure 2.

The module with InGaP/GaAs two-junction cell on Si solar cell was provided by SHARP and tested at the University of Miyazaki simultaneously with a 2T three-junction module (InGaP/GaAs/InGaAs). An I-V curve tracer measured the performance of these two PV modules. Pyranometers were placed at 25° (EKO MS-602) and 35° (EKO MS-411) slope angles to measure global irradiance. Another pyranometer was installed on a two-axis sun-tracking (EKO MS-602) to measure global normal irradiance. Every 3 min, the irradiance of these sensors was captured and logged from 5:30 a.m. to 6:30 p.m. The solar spectrum, specifically the global spectrum on the sloped surface, was measured by the spectro-radiometers (EKO MS-711, MS-712). Their slope angles were 35°. Measurements of the global spectrum were every 10 min from 5:00 a.m. to 8:00 p.m.

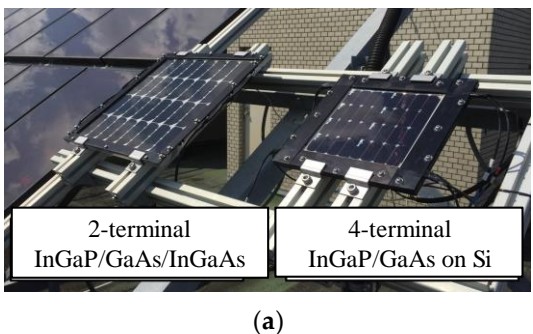
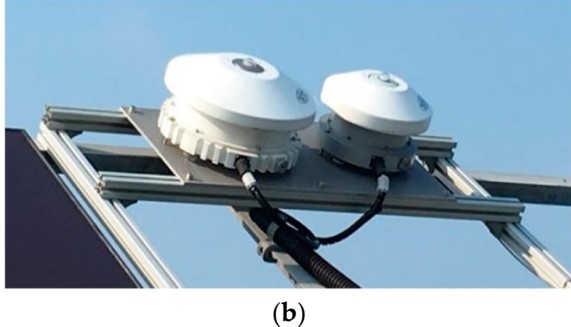

(**a**)                                                                 (**b**)

**Figure 2.** Measured 4T III-V/Si module and its solar cell structure: (**a**) Description of measured 4T III-V/Si module, 2 + 1 junctions (right) with comparison to the 2-terminal III-V 3-junction module. Note that the number of junctions is the same; (**b**) Spectro-radiometers [97].

### 2.3. Spectrum Model

The performance model we used was identical to the one we used to analyze the PV module using the 2T configuration [97]. We called it the MS2E model (Miyazaki Spectrum-to-Energy method). Let us describe a rough flow of the analysis and the model.

First, we need to define the solar spectrum. The spectrum we assumed was a linear combination of the spectrum of the clear sky condition and that of the overcasting state [97]. For the spectrum of the overcasting state it was assumed that the solar power was lost just by absorption, corrected by the solid angle of the sky influenced by the slope angle of the module. Then, the global solar spectrum is approximated as Equations (1) and (2).

$$I_\lambda = f I_{1\lambda} + (1-f) I_{2\lambda} \tag{1}$$

$$f = \frac{DNI}{\int_{300\ nm}^{4000\ nm} I_{d\lambda} d\lambda} \tag{2}$$

where, $I_\lambda$ is the global spectral irradiance calculated by our spectrum model covering all weather at a wavelength. The unit of $I_\lambda$ is W/m²nm. $f$ is the weather correction factor defined by Equation (2), $I_{1\lambda}$ is the global spectral irradiance calculated using Bird's spectrum model [96] at a wavelength. The unit of $f$ is W/m²nm. $I_{2\lambda}$ is the global spectral irradiance calculated by a spectrum model assuming full cloud covering the sky at a wavelength. The unit of $I_{2\lambda}$ is W/m²nm. $DNI$ is the direct normal irradiance, and $I_{d\lambda}$ is the direct normal solar spectral at a wavelength. The unit of $I_{d\lambda}$ is W/m²nm. The spectrum calculated by this model, in contrast to Bird's model, is shown in Figure 3 [97]. Note that the Bird's spectrum model was improved by considering atmospheric parameter variability. The Y-axis corresponds to the normalized global spectrum irradiance by integrated spectral irradiance. The black

trend line is the measured and normalized global solar spectral irradiance. The gray trend line is the reference spectrum in AM 1.5G. The red trend line and the blue trend line are the calculated global solar spectral irradiance by the MS2E method. Note that Bird's model only considers air mass, namely, the atmospheric parameters are constant. In the wintertime, atmospheric parameters are close to those under the standard conditions, so that the estimated solar spectrum approaches to the reference AM1.5G spectrum. In the summertime, the aerosol density often drops lower than that of the standard value, and the precipitable water grows larger. The short-wavelength region of the solar spectrum becomes thick, and the long-wavelength part becomes thin. During cloudy days, the influence of cloud appears in the short-wavelength part of the solar spectrum so that the long-wavelength region drops.

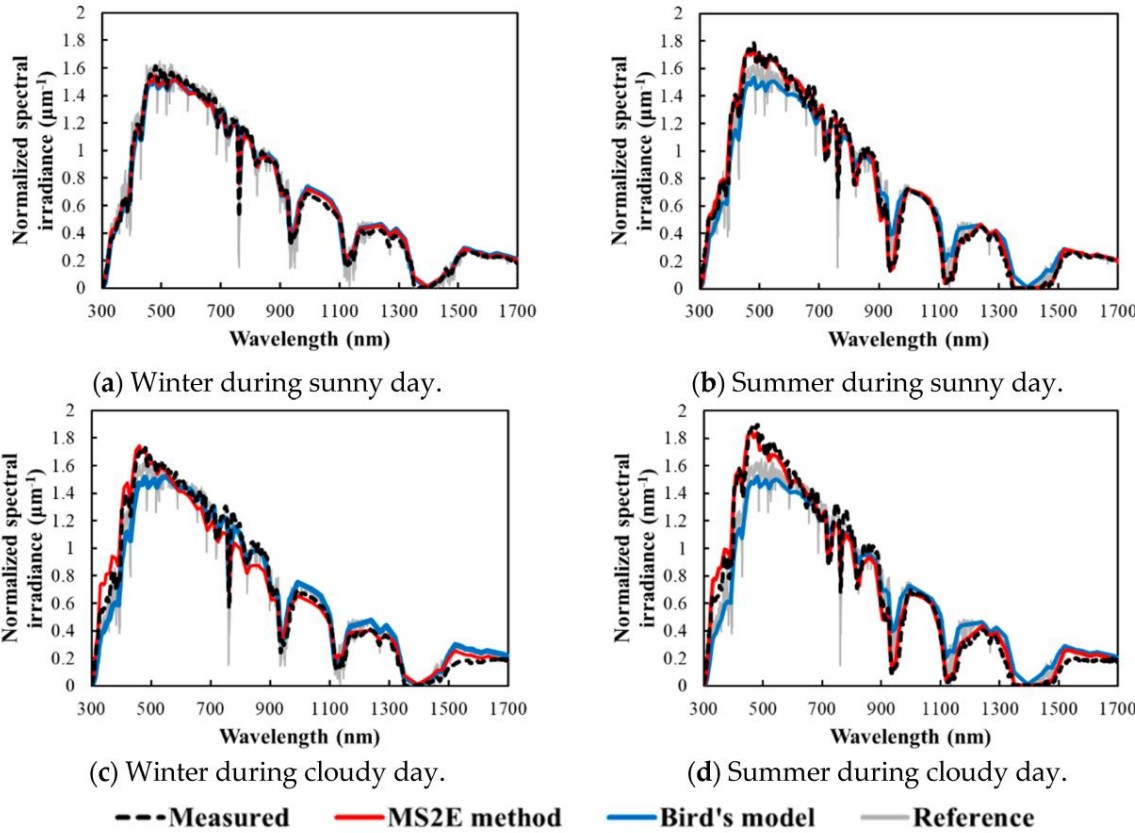

**Figure 3.** Comparison with measured and estimated values (Bird's model and Miyazaki Spectrum-to-Energy (MS2E) model) of global irradiances tilted at 35° [97]: (**a**) Solar spectrum in the sunny day in winter; (**b**) Solar spectrum in the sunny day in summer; (**c**) Solar spectrum in the cloudy day in winter; (**d**) Solar spectrum in the cloudy day in summer.

Figure 4 indicates the measured aerosol density and precipitable water by the curve fitting to the measured solar spectrum. The aerosol optical depth was much lower than the standard value used to the calculation of the reference spectrum in summer (Figure 4a). This is why the measured and estimated irradiances in the summertime are higher than those of global irradiances (AM 1.5G) at the wavelength range of 450–550 nm. The precipitable water was much higher than the standard value used for the calculation of the reference spectrum in summer (Figure 4b). This is why the measured and estimated irradiance dips are broader than those of global irradiances (AM 1.5G) at the wavelength range of 1350–1450 nm and around 1150 nm.

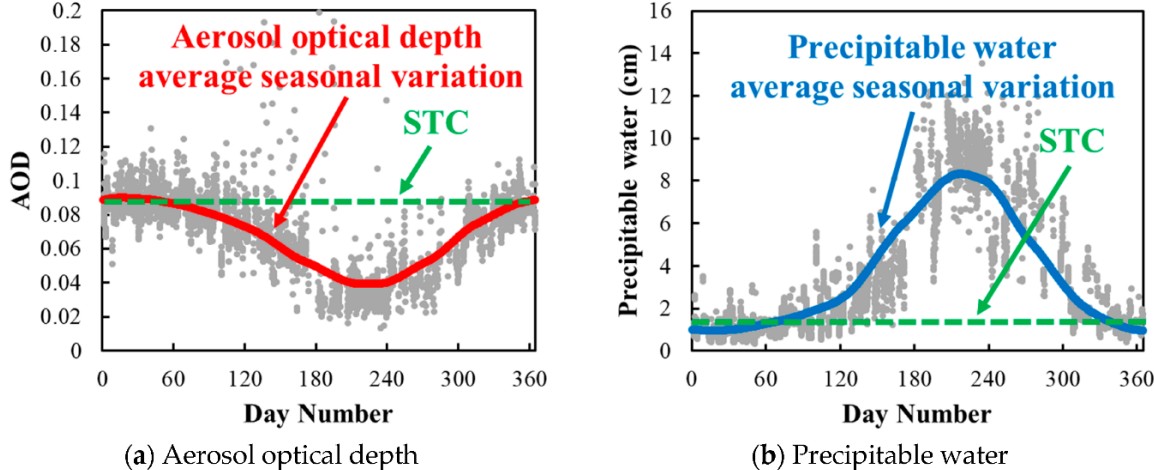

(**a**) Aerosol optical depth       (**b**) Precipitable water

**Figure 4.** Seasonal trend of atmospheric parameters in Miyazaki [97]: (**a**) Seasonal trend of aerosol optical depth; (**b**) Seasonal trend of precipitable water.

### 2.4. Performance Model

The power output of the module is the product of the short-circuit current, open-circuit voltage, and fill factor (*FF*). First, *FF* was calculated by the ratio of the spectrum mismatching. The calculation step is as follows. First, generating a correlation chart between calculated *FF* and the ratio of mismatching at first using random numbers. Then, the trend curve of these two parameters was fitted to the parabolic curve. Finally, the *FF* was represented as the function of the spectrum-mismatching ratio. The short-circuit current can be calculated as the integral of the product of the spectral irradiance and spectral efficiency of the module, which can be derived from the external quantum efficiency affected by spectrum mismatching. The angular characteristics in the photon absorption were measured in advance. The detailed calculation procedure was identical to our previous work [97].

We also needed to consider the coupling between spectrum and angles. The atmospheric parameters are dependent on each other. For example, a combination of the incident angle modifiers and the orientations results in a diverse mixture of the direct and the diffused sunlight. The atmospheric parameters were calculated by the spectrum, by a data-fitting calculation called the Bird's model [98,99] at the University of Miyazaki [100]. The model for this analysis is given in Figure 5 [1]. The nonlinear effect from distributed effects often seen in III-V (especially concentrator cells) was not considered [101–103].

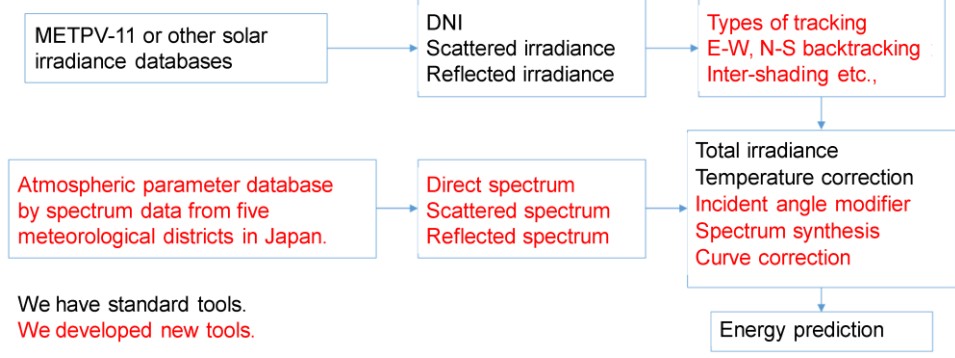

**Figure 5.** The performance model of the tandem solar cells, considering the spectrum and angle interaction [1].

## 3. Results

First, we examined our performance model (MS2E model) that could be applied to the PV module using 4T tandem cells. We monitored the module performance in every 3 min from 05:30 to 18:30 on 3 January 2019. The slope angle was 35°. The location of the test site at the University of Miyazaki was N 31.83°, E 131.42°. The result of the measurement and the validation result is shown in Figure 6 and Table 2. The output trend in Figure 6 was decomposed to InGaP/GaAs top cell and Si bottom cell, namely, we applied independent MPPT (maximum power point tracking) search to each pair of the terminals. The predicted output of the 4T tandem module using our MS2E performance model supported by the all-climate spectrum model (Equations (1) and (2)) matched well. It was also shown that the accuracy in prediction of the daily energy yield is within plus or minus 2% of error.

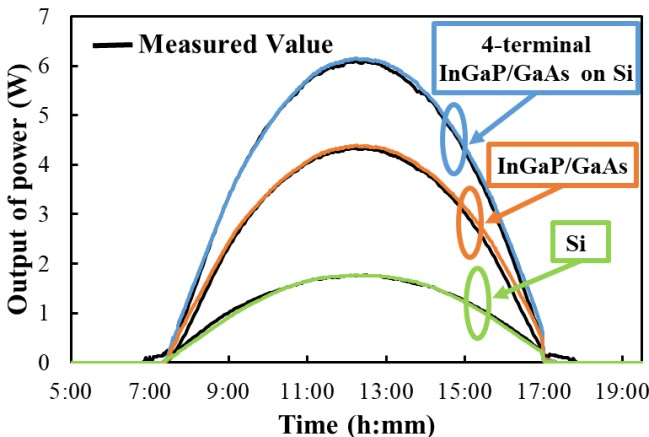

**Figure 6.** Validation of the performance model (MS2E model) to 4T tandem module, decomposing performance in each output terminal (InGaP/GaAs top cell and Si bottom cell).

**Table 2.** Comparison between measured and estimated (predicted by MS2E model) energy yield in a day (3 January 2019).

|  | Measured | Estimated | Error |
| --- | --- | --- | --- |
| InGaP/GaAs on Si [1] | 39.6 Wh | 40.4 Wh | 1.9% |
| InGaP/GaAs | 28.0 Wh | 28.9 Wh | 3.4% |
| Si | 11.6 Wh | 11.4 Wh | 1.9% |

[1] 4T configuration independently takes the output from the top and bottom cells; the production of the InGaP/GaAs on Si module is a simple sum from the InGaP/GaAs top layer and Si bottom layer.

## 4. Discussion

It was shown that our MS2E model is sufficiently accurate to discuss the annual and outdoor performance of 4T tandem modules. Next, let us discuss and compare with other types of tandem modules with consideration of regional variations.

### 4.1. Comparison Between 4T and 2T Configuration

The annual performance of the 4T tandem module was calculated by the validated MS2E model. For comparison of robustness to the seasonal spectrum change, it is essential to introduce a normalized scale of the performance, because the area, nominal output, cell type, and power conversion efficiency are different. We used performance ratio *PR* and the ratio of the performance peak-to-peak $R_{pp}$, which are defined by Equations (3) and (4).

$$PR = \frac{(Outdoor\ efficiency)}{(Efficiency\ measured\ by\ STC)} \tag{3}$$

$$R_{pp} = 1 - \frac{(peak\ to\ peak\ of\ PR\ in\ 4T)}{(peak\ to\ peak\ of\ PR\ in\ 2T)} \tag{4}$$

Note that STC means the standard testing condition. The *PR* value given by Equation (3) corresponds to how much power generation performance of the photovoltaic module drops in outdoor operation about the indoor testing result measured by the standard testing condition. The $R_{pp}$ value given by Equation (4) corresponds to the degree of suppression of the seasonal variation of performance of 4T configuration relative to standard 2T configuration. Both *PR* and $R_{pp}$ values were integrated throughout the day, and their daily trend was plotted in time series. The result with contrast to 2T configuration is shown in Figure 6 and Table 3. Note that the peak-to-peak value of the *PR* variation of 4T configuration was 0.084, and that of 2T was 0.139. The degree of improvement of seasonal variation $R_{pp}$ was 39.8% (Figure 7a). As a result, the annual energy yield per nominal power of 4T configuration increased to 1500 kWh/kW, and that of 2T was 1442 kWh/kW (Table 3). The improvement was mainly seen in the summer. It corresponded to the variation of water precipitation (Figure 7b). The bandgap of the bottom cell of Si is 1.11 eV, and the absorption edge is 1100 nm so that the performance of the Si bottom cell would not be affected by the water absorption typically seen in around 1200 nm.

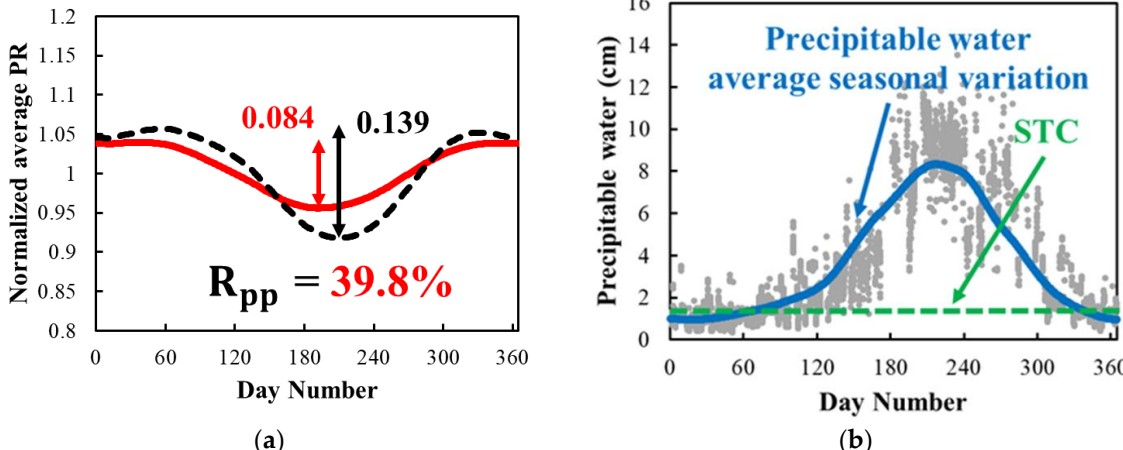

(a)

(b)

**Figure 7.** Comparison of the annual output between 4T configuration and 2T configuration affected by the variation of the atmospheric parameter: (**a**) Predicted seasonal fluctuation of the normalized energy yield of 4T (red and solid line) and 2T (black and dashed line) configuration; (**b**) Seasonal variation of precipitable water (optical depth) in our measurement that is likely to be responsible for the difference of behavior between 2T and 4T configuration. Note that this chart is identical to Figure 4b [99].

**Table 3.** Summary of the outdoor performance of 4T on-Si tandem solar cell and normal 2T three-junction tandem solar cell.

| | PR Peak-to-Peak Value | Annual Energy Yields (kWh/kW) |
|---|---|---|
| InGaP/GaAs on Si [1] (4-terminal configuration) | 0.084 | 1500 |
| InGaP/GaAs/InGaAs (2-terminal configuration) | 0.139 | 1442 |

[1] 4T configuration independently takes the output from the top and bottom cells; the production of the InGaP/GaAs on Si module is a simple sum from the InGaP/GaAs top layer and Si bottom layer.

On the other hand, the InGaAs bottom cell with 1.0 eV bandgap is affected by the absorption of this band. In 2T configuration, the reduction of absorbed photons in this water band constrains the current output of series-connected entire junctions. In addition to the difference in numbers of terminals (4T or 2T), the difference of the bandgap of the bottom cell affected the degree of the seasonal variation.

### 4.2. Regional Difference in the Behavior of 4T and 2T Performance

The fact that the annual change is affected by water precipitation implies that the gain by 4T configuration may be strongly influenced by the local climate.

With the validation of the MS2E model in the estimation of PV output, we applied it more broadly. The solar database METPV-11 [104,105] has a solar irradiance measurement dataset for more than 800 sites in Japan. We examined the seasonal energy yields in various locations with different climates. For the determination of the atmospheric parameters, measurement data of the solar spectrum are needed. NEDO (New Energy and Industrial Technology Development Organization) classified Japan into five different solar radiation climate zones to clarify regional differences in solar irradiation conditions. The global solar spectrum was monitored at five sites, including Naganuma (subarctic zone), Tosu (temperate zone), and Okinoerabu (subtropical zone). We extracted aerosol density and water precipitation in each solar radiation climate zone. Correctly, the aerosol density and water precipitation were derived using a model in Equations (1) and (2) that minimizes the deviation of the global spectrum between the experimental values measured and the model values. Note that the estimated global spectrum is a function of the variables of aerosol density and water precipitation [97].

The result is shown in Figure 8. From the left to the right column, with the transition from the subarctic zone, to the temperate zone, and subtropical zone, water precipitation, which was calculated by optical absorption of the spectrum using the method mentioned above, increased. The gain of 4T configuration ($R_{pp}$ value) increased accordingly.

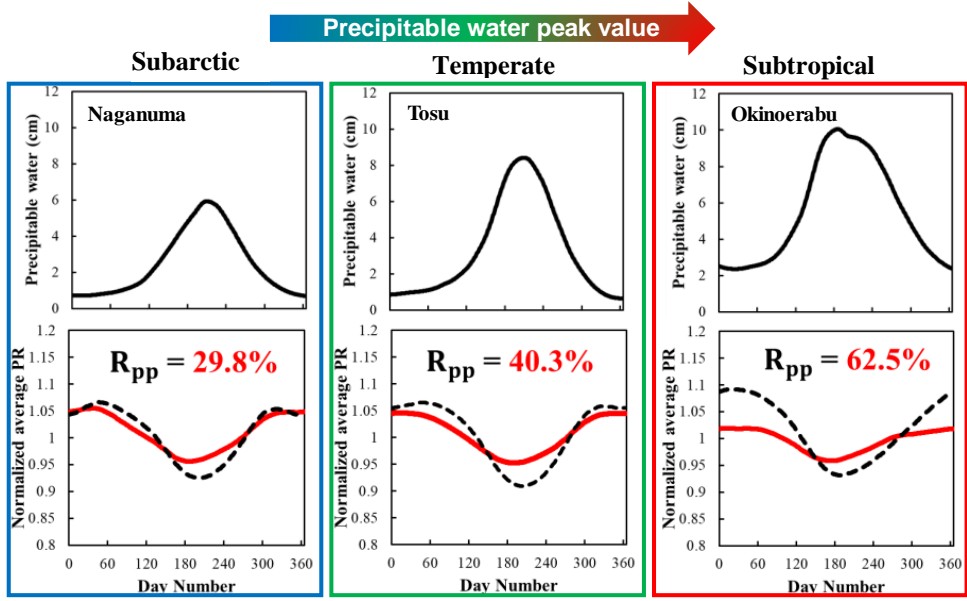

**Figure 8.** Comparison of the annual output between 4T configuration and 2T configuration affected by the variation of the atmospheric parameter. The red solid trend line in the bottom charts corresponds to the normalized energy yield of 4T, and the black dashed trend line corresponds to that of 2T configuration.

### 4.3. Further Performance Improvement

Although the 4T configuration effectively improves the seasonal loss of multi-junction modules, there still is some annual drop. It is because the top cell has two junctions, and its spectrum mismatching drops the output of the top cell.

Further improvement of the mismatching was proposed in several articles. One is fine-tuning of the bandgap energy [106]. The second is enhancing radiative coupling that was shown in the concentrator PV application [107] and non-concentrating application [1].

## 5. Conclusions

Tandem solar cells are highly efficient, and various types of the device structure were studied. On-Si tandem is one of them, and it has a distinct advantage of cost, using well-established Si solar cell technology. Regardless of the type of material, the annual performance of the tandem solar cells does not perform well due to the spectrum mismatching loss. The modeling of the spectrum mismatching loss was studied relying on the airmass variation. The intensive study on CPV performance in more than 20 years revealed that the fluctuation of atmospheric parameters played an essential role. Due to the development of the new application of the high-efficiency solar cell, including vehicle-integrated solar cells, the precise annual energy yield modeling of the tandem solar cells is required. The knowledge of precise spectrum-mismatching modeling in CPV is expanded to the non-concentration standard installation. 4T on Si tandem solar cell is a good candidate for robustness to spectrum variation. Its outdoor operation and energy yield modeling was intensively studied in this article. The model did not rely only on airmass but considered real fluctuation of the spectrum in all kinds of climates considering atmospheric fluctuation.

Many previous studies tried to calculate the spectrum conditions only by airmass and also tried to apply the model that was developed for clear-sky conditions. The airmass-based calculation was proved inaccurate in the intensive investigation of CPV modules using three-junction tandem cells. The lack of an all-climate spectrum model was not the issue for the application to CPV, because it only used the direct sunlight. However, for the application to the standard installation (non-concentration), we must count the energy generation on the partially clear-sky days and cloudy days. A new idea of approximation of the all-climate spectrum model was developed and tried in this study, and it successfully modeled the energy yield of the tandem cells in all climates.

The advantage of the 4T configuration of the photovoltaic module using a triple-junction solar cell, precisely, InGaP/GaAs on Si solar cell, was compared with a conventional 2T triple-junction module, specifically, InGaP/GaAs/InGaAs solar cell. The behavior of seasonal variation of performance and spectrum influence was modeled and validated by outdoor measurement.

The annual amplitude of the seasonal peak-to-peak performance ratio improved by about 40%. This robust performance of the seasonal fluctuation of the solar spectrum is useful to applications to high-performance photovoltaic applications like zero-emission buildings and light-weight aerospace applications [108]. For the use in vehicle-integrated photovoltaic systems, it is also essential to consider 3D solar irradiance (modeling [109] and measurement [110]), as well as the performance of the photovoltaic unit on a curved surface (modeling [111,112] and the standard [113]).

The seasonal fluctuation of the 2-terminal triple-junction solar is also responsible for seasonal variation of the water precipitation. The above-validated model was expanded to various climate zones. The advantage of 4T on Si configuration increases (subarctic zone < temperate zone < subtropical zone).

It is important to note that the model we developed and validated is not only applied to 4T InGaP/GaAs on Si solar cell, but can be extended to every type of the tandem solar cells in principle, including Perovskite on Si, Perovskite on CIGS, Polymer tandem, and III-V tandem, regardless of monolithic, wafer-bonding, and mechanical stack. It can even be applied to partial concentrator tandem cells [77] and super-multi-junction solar cells [1].

**Author Contributions:** Conceptualization: Y.O., and K.A.; methodology: H.T., H.S., Y.O., and K.A.; validation: H.T., H.S., Y.O., and K.A.; formal analysis: H.T., H.S., and Y.O.; data curation, H.T., H.S., and Y.O.; writing—original draft preparation, H.T., K.A. and Y.O.; writing—review and editing: H.T., Y.O., and K.A.; supervision: K.N. and M.Y. All authors have read and agreed to the published version of the manuscript.

**Funding:** This research was funded by the New Energy and Industrial Technology Development Organization (NEDO), and a grant for Scientific Research on Priority Areas from the University of Miyazaki.

**Acknowledgments:** This work was supported in part by the New Energy and Industrial Technology Development Organization (NEDO).

**Conflicts of Interest:** The authors declare that there are no conflicts of interest.

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
