# Peer review of "The Outdoor Field Test and Energy Yield Model of the Four-Terminal on Si Tandem PV Module"

_applsci, doi:10.3390/app10072529_

Round 1

Reviewer 1 Report

Overall Comment:

This paper presents the result of the performance analysis of a four terminal on Si Tandem PV module.
Authors use the performance model of a previous research of theirs.
The scientific novelty of the approach is not relevant therefore it becomes preminent that:
1) authors stress out what are the motivations with respect to other papers
2) and what are the original elements of this research with respect to other related researches.

Nevertheless, the paper is fluent and easy to read.

Below some technical comments:

Introduction:
In the introduction authors should provide more details about the motivation behind this research paper as, in the current form, the introduction is mainly dedicated to resume the efficiency results of other Si solar panels from related papers.

Few typos are higlighted:
Line 38: HIgher = Higher
Line 133: pf = of

Section 2.2: What do authors mean with this sentence:
"The monitoring duration was 3 min, from 5:30 a.m. to 6:30 p.m."

Conclusions:
Likewise the introduction, conclusions are poor. Conclusions is one of the most important section as it might resume the motivations of the research, the results achieved (which is the only element present at the moment), a discussion that explains the results presented and future research goals which, in this case, cannot be limited to a phrase (243-245) that looks random as it is placed there.

Author Response

Response to reviewer 1

This paper presents the result of the performance analysis of a four terminal on Si Tandem PV module.

Authors use the performance model of a previous research of theirs.

The scientific novelty of the approach is not relevant therefore it becomes preminent that:

1) authors stress out what are the motivations with respect to other papers

2) and what are the original elements of this research with respect to other related researches.

In the introduction authors should provide more details about the motivation behind this research paper as, in the current form, the introduction is mainly dedicated to resume the efficiency results of other Si solar panels from related papers.

Likewise the introduction, conclusions are poor. Conclusions is one of the most important section as it might resume the motivations of the research, the results achieved (which is the only element present at the moment), a discussion that explains the results presented and future research goals which, in this case, cannot be limited to a phrase (243-245) that looks random as it is placed there.

Author’s response

I appreciate your suggestion. Now, both the introduction section and conclusion section are entirely updated.

Introduction section:

High-efficiency is the typical research target of photovoltaic technology. However, it is also known through field experience and theoretical analysis considering spectrum fluctuation that the photovoltaic system that wins the race of efficiency does not always gain the best yield in the real installation [1-4].

Currently, Si solar cell has commonly prevailed in the market. The best efficiency of the Si that was confirmed testing laboratories is 26.7% [5], and the theoretical limit is 29.43% [6]. For further improvement of efficiency, multi-junction or tandem configuration is preferred. The idea of multi-junction cells was suggested [7] and investigated [8] in the early days of photovoltaic technology. The significant progress was triggered by AlGaAs/GaAs tandem cells with tunnel junctions [9] and metal interconnections [10-12]. At that moment, it was predicted that the power-conversion efficiency of multi-junction solar cells would reach close to 30% [13], but this was not fulfilled by difficulties in stable tunnel junctions [14] and defects in the AlGaAs [15]. The break-through was a high-performance, stable tunnel junctions with a double-heterostructure [16]. InGaP was introduced for the top cell [17], and as a result, it was finally achieved to almost 30 % efficiency by a  GaInP/GaAs cell [18]. Higher efficiencies have been made with InGaP/GaAs/InGaAs triple-junction cells [19] and with a 5-junction cell [20].

On-Si tandem solar cells use the widely-used Si solar cell as the bottom junction of the tandem solar cell. Because the technology of Si solar cells is well-established, the production cost of the solar cell is expected to reduce, at least, the one to the substrate or the bottom junction [21]. The III-V/Si (III-V on Si) 3-junction and 2-junction tandem solar cells right now exhibit excellent efficiency with 35.9% [22] and 32.8% [22]. The perovskite on Si 2-junction tandem solar cells reached to 28.0% [23]. That of CdZnTe on Si tandem solar cell was 16.8% [24], and GaAs nano-wire on Si tandem solar cell was 11.4% [25]. The III-V/Si tandem solar cell has the best efficiency among them. Related to this III-V/Si tandem solar cell technologies, several module technologies, including a partial concentrator module [26]. The tested power conversion efficiency of a pair of InGaP/GaAs partial concentrator cell on Si, including optical loss of the concentrator optics, was 27.1% [27]. The partial concentrator module, also using InGaP/GaAs partial concentrator cell on Si and designed for automobile application, showed 21.5% module efficiency [28]. The non-concentrating module using InGaP/GaAs cell on InGaAs by 4T configuration showed 31.17% [1, 29].

Another big group of the candidates of the on Si tandem cells is Perovskite on Si tandem cell. Including a review of the 4T Perovskite on Si cell [30], and challenging on the growth on the textured silicon [31-33]. Related to the topic of this paper, consideration of the performance of the Perovskite on Si tandem cell was done extending the standard testing condition (AM1.5G spectrum and consideration of the diffused sunlight). Still, it was not validated by the annual outdoor operation [34].

Regardless of the material type combined with Si bottom junction, mismatching loss by the spectrum change will be a big issue [35-40]. The intensive studies of the spectral sensitivity of the tandem cell were firstly done in the analysis of the concentrator photovoltaic (CPV) since tandem solar cells were expensive. Still, high-performance and the concentration operation was commercially advantageous to the application [41-50]. The spectrum shift from the standard AM1.5G spectrum destroys the balance of the output electrical current from the sub-cell. Then, the sub-cell with the least output current constrains the total electrical current by the conservation of the carrier or the Kirchhoff’s law. It is called “spectrum-mismatching loss.” It is an inherent loss for every type of tandem solar cell, regardless of CPV or standard flat-plate application, irrespective of on-Si or on-CIGS tandem, or regardless of III-V or Perovskite tandem solar cells except for more than 3 terminal configurations where the output of the sub-cells is individually connected to the load. It is essential to note that a variation of the solar spectrum by sun height and fluctuation of scattering and absorption of the air by the seasonal effect will affect the annual performance. However, many pieces of research and developments were done to minimize the influence of spectrum change by the improvement of the solar cell design [51-56].

Research and development seeking for the device design with the robustness to spectrum change have been made in the past 20 years, including a computer model named Syracuse by Imperial College London [2, 57-58]. For CPV applications that have been the center of the study of the tandem cells with the robustness of spectrum issues, it was understood that the chromatic aberration of the concentrator optics enhanced the spectrum-mismatching loss [59-63]. However, such loss could be solved by the innovation of optics, including homogenizers and the secondary optical element (SOE) [64-65]. The remaining problems of the spectrum-mismatching loss have been overcome by the adjustment of the spectral response of each sub-cell, including overlapping the absorption spectrum and broadening the absorption band to the zone of massive fluctuation.

The model, theory, and advanced design of CPV, as well as innovations for solving the issue of the spectrum loss in the real-world solar irradiance, were tried to be validated by the outdoor energy yield monitoring. However, it was not very successful. One possible reason was the performance of CPV was also sensitively affected by the tracking error and misalignment of the concentrator optics and the solar cell. However, this issue was solved by the series of innovations of the testing method of the misalignment [66], validation in the production line of the CPV module [67], and the analysis of the impact on the energy output [68-69]. After solving uncertain factors, the influence of the spectrum change was re-examined. However, it was found that the standard model using air-mass (or the sun height) as the parameter of the solar spectrum, which has been used by many scientists and for more than 20 years, could not explain the seasonal trend of the performance by analysis [70] and the same results using variance (ANOVA) [71]. The new-findings that directly and more strongly influence the spectrum than the air-mass was a fluctuation of atmospheric parameters. This was because the impact by the sun-height appears strongly when the airmass value is more than 2. And both summer and winter have the early morning and the late evening time equally. Namely, without consideration of atmospheric parameters, it is next to impossible to predict the annual energy yield, in other words, it is next to impossible to design the optimal tandem solar cells in yearly energy yield.

Recently, tandem solar cells have been considered for use in non-concentrating applications, out of CPV, including vehicle-integrated photovoltaic (VIPV) and car-roof photovoltaics (PV) [72-84]. It was thought that most electric vehicles (EV) might be able to run by solar energy using tandem cells on the car roof [1]. The area of the car roof and other car-bodies are limited. Moreover, solar cells cannot be laminated to an undevelopable curved surface of the car body. It is difficult to cover the restricted area of the car-roof surface entirely. Therefore, extremely high performance that can be brought by tandem solar cells is highly required.

The analysis of the spectrum sensitivity on CPV was done in our previous research [85-86]. The calculation and analysis for CPV were more straightforward because we did not have to consider angular effects combined with the mixture of the direct and diffused components of the sunlight. Different from CPV applications that the cell is always perpendiculary illuminated by the sun using a solar tracker, and only utilizes the direct sunlight, the typical application needs to use a diffused component of the sunlight from the sky, ground reflection, and skewed solar rays, with a combination of direct and diffused elements as a function of the sun orientation relative to the solar panel orientation. For an extension to non-concentrating applications, we need to solve the complicated coupling of spectrum and angles (Table 1). The key parameters are atmospheric parameters that are dependent on each other. For example, different incident angle modifier and different orientation lead to a diverse mixture of direct and diffused sunlight.

Table 1. The difference in performance modeling between CPV and standard installation [1]. The outdoor performance of CPV was intensively studied in more than 20 years considering spectrum mismatching problem of the tandem solar cells.

CPV 1

Normal Installation

Solar spectrum

Only direct

A mixture of direct, diffused from the sky, and reflection

Angle

Always normal

Varies by time and seasons

Spectrum by angle

Constant (only normal)                

Needs consider coupling to angle

1 It only generates power only by direct solar irradiance using a 2-axis solar tracker.

Four-terminal (4T) tandem cells were designed so that the output of the Si cell and other top junctions were taken independently, thus robust to the spectrum change. In the case of III-V/Si three-junction solar cells that are frequently considered as an excellent candidate for the high-efficiency solar cells, the output terminal from the top cell comes from two-junction III-V solar cells, and the top two-junction cell is still susceptible to spectrum change. Therefore, a long-term field test of the module using III-V/Si 4T solar cells is essential to the validation of the use of this configuration. Currently, the best efficiency confirmed by the third-party is 33.3% [87].

Other groups of the 4T type tandem cells are partial concentrator cells and 3T tandem cells. The partial concentrator solar cell was first proposed in 2017, using wide-acceptance concentrator optics that selectively concentrate onto the top III-V cells stacked on the larger size Si bottom cell. This type of the tandem solar cell was considered to the application to VIPV [89] and various design method and modeling were invented [90-92]. Several prototypes were made [79] and as high as 27.4 % of annual average efficiency was anticipated [78]. 3T type tandem cells are basically connecting one of the terminals of the pairs of two-terminal. This method has an advantage of simplifying the interconnection of the solar cells among the PV module. Several types of the tandem cells, including Ge-based III-V cell [93], polymer type [94], III-V monolithic [95], and III-V on Si [96]. It was also studied advantage of the better spectrum matching operation using this 3T configuration [97].

The background of this study and our motivation are summarized as follows:

  1. Tandem solar cells are high-efficiency, and various types of the device structure were studied. On-Si tandem is one of them and it has a distinct advantage of the cost using well-established Si solar cell technology.
  2. Regardless of the type of the materials, the annual performance of the tandem solar cells does not perform well by the spectrum mismatching loss.
  3. The modeling of the spectrum mismatching loss was studied relying on the airmass variation. The intensive study on CPV performance in more than 20 years revealed that the fluctuation of atmospheric parameters played essential role.
  4. Due to the development of the new application of the high-efficiency solar cell, including vehicle-integrated solar cells, the precise annual energy yield modeling of the tandem solar cells are demanded. The knowledge in precise spectrum-mismatching modeling in CPV is expanded to the non-concentration standard installation.
  5. 4T on Si tandem solar cell is a good candidate for the robustness to the spectrum variation. Its outdoor operation and energy yield modeling was intensively studied in this article. The model did not rely only on airmass, but considered real fluctuation of the spectrum in all kind of climate considering atmospheric fluctuation.

Conclusion section:

Tandem solar cells are high-efficiency, and various types of the device structure were studied. On-Si tandem is one of them and it has a distinct advantage of the cost using well-established Si solar cell technology. Regardless of the type of the materials, the annual performance of the tandem solar cells does not perform well by the spectrum mismatching loss. The modeling of the spectrum mismatching loss was studied relying on the airmass variation. The intensive study on CPV performance in more than 20 years revealed that the fluctuation of atmospheric parameters played essential role. Due to the development of the new application of the high-efficiency solar cell, including vehicle-integrated solar cells, the precise annual energy yield modeling of the tandem solar cells are demanded. The knowledge in precise spectrum-mismatching modeling in CPV is expanded to the non-concentration standard installation. 4T on Si tandem solar cell is a good candidate for the robustness to the spectrum variation. Its outdoor operation and energy yield modeling was intensively studied in this article. The model did not rely only on airmass, but considered real fluctuation of the spectrum in all kind of climate considering atmospheric fluctuation.

Many previous studies tried to calculate the spectrum conditions only by airmass and also tried to apply the model that was developed for the clear-sky conditions. The airmass based calculation was proved inaccurate in the intensive investigation to CPV modules using three-junction tandem cells. The lack of all-climate spectrum model was not the issue for the application to CPV, because it only used the direct sunlight. However, for the application to the standard installation (non-concentration), we must count the energy generation in the partially clear-sky days and cloudy days. A new idea of approximation of the all-climate spectrum model was developed and tried to this study and successfully modeled the energy yield of the tandem cells in all climates.

The advantage of the 4T configuration of the photovoltaic module using a triple-junction solar cell, precisely, InGaP/GaAs on Si solar cell, was compared by conventional 2T triple-junction module, specifically, InGaP/GaAs/InGaAs solar cell. The behavior of seasonal variation of performance and spectrum influence was modeled and validated outdoor measurement.

The annual amplitude of the seasonal peak-to-peak performance ratio improved by about 40 %. This robust performance of the seasonal fluctuation of the solar spectrum is useful to applications to high-performance photovoltaic applications like zero-emission buildings and light-weight aerospace [110]. For the use to the vehicle-integrated photovoltaic, it is also essential to consider 3D Solar Irradiance (modeling [111] and measurement [112]), as well as the performance of the photovoltaic in the curved surface (modeling [113] and the standard [114]).

The seasonal fluctuation of the 2-terminal triple-junction solar is also responsible for seasonal variation of the water precipitation. The above-validated model was expanded to various climate zones. The advantage of 4T on Si configuration increases (subarctic zone) < (temperate zone) < (subtropical zone).

It is important to note that the model we developed and validated is not only applied 4T InGaP/GaAs on Si solar cell, but can be extended to every type of the tandem solar cells in principle, including Perovskite on Si, Perovskite on CIGS, Polymer tandem, III-V tandem regardless of monolithic, wafer-bonding and mechanical stack. It even can be applied the partial concentrator tandem cells [78] and the super-multi-junction solar cells [1].

Few typos are higlighted:

Line 38: HIgher = Higher

Line 133: pf = of

Author’s response

I appreciate your suggestion. The entire text was corrected. English was polished as well.

Section 2.2: What do authors mean with this sentence:

"The monitoring duration was 3 min, from 5:30 a.m. to 6:30 p.m."

Author’s response

I appreciate your suggestion. It was corrected as “Every 3 minutes, the irradiance of these sensors was captured and logged from 5:30 a.m. to 6:30 p.m.”

Reviewer 2 Report

In this manuscript the authors study the performance of tandem solar cells based on InGaP/GaAs and Silicon solar cells. The main objective of the study is to establish which geometry (2 Terminal versus 4 Terminal) is the best option for the use of solar cells in the automotive application.

First author belong to the car manufacture company Toyota, and this manuscript is a new contribution in a long and fruithful series of publications from the same group in this specific technological application. 

The work is well described, solar tandem cells are very well characterized in terms of incident angle, forecasting and precitable water among other parameters. Conclusions are solid.

My only concern about this study is that the number of interested readers could be limited to this specific application. However, considering the high quality of the work I have no hesitation in recommend its publication in Applied Sciences journal.  

Author Response

Response to reviewer 2

My only concern about this study is that the number of interested readers could be limited to this specific application. However, considering the high quality of the work I have no hesitation in recommend its publication in Applied Sciences journal. 

Response by authors

I appreciate your suggestion. I updated abstract, introduction, and conclusion with additional explanation in the main body so that this article is not the one for III-V on Si tandem solar cell, but can be potentially applied to entire types of the tandem solar cells. English was also polished.

Reviewer 3 Report

This paper reports the outdoor field test energy yield model of the four-terminal III-V cells on Si tandem photovoltaic module. The 4-terminal III-V on Si tandem photovoltaic module had about 40 % advantage in seasonal performance loss compared with standard InGaP/GaAs/InGaAs 2-terminal tandem photovoltaic module. This study is interested to the readers. However, the following issues should be addressed:

  1. The keyword of “output forecasting”, “aerosol optical depth”, “precipitable water” and “incident angle” are not suitable, because they are not present in Title and Abstract of the manuscript.
  2. Page 3 of 11, Line 109-112: the authors claimed that “Il1 is the global spectral irradiance calculated using Bird’s spectrum model [37] at a wavelength (W/m2nm), Il2 is the global spectral irradiance calculated by a spectrum model assuming full cloud covering the sky at a wavelength (W/m2nm)”. The description of “at a wavelength (W/m2nm)” is very confused to readers. The authors should be revised and give the information at what wavelength is.
  3. Page 4 of 11, Fig. 3: the measured and estimated irradiances in the summertime are high than that of global irradiances (AM 1.5G) at the wavelength range of 450-550 nm. Please give the reasons that the irradiances are higher at this wavelength range.

Author Response

Response to reviewer 3.

I appreciate your intensive review of improving the quality of our article.

The keyword of “output forecasting”, “aerosol optical depth”, “precipitable water” and “incident angle” are not suitable, because they are not present in Title and Abstract of the manuscript.

Authors response:

I appreciate your suggestion. The list of keywords is updated. The new train of the keyword is;

photovoltaic; solar spectrum; tandem cell; energy yield model; on-Si tandem; terminal

Page 3 of 11, Line 109-112: the authors claimed that “Il1 is the global spectral irradiance calculated using Bird’s spectrum model [37] at a wavelength (W/m2nm), Il2 is the global spectral irradiance calculated by a spectrum model assuming full cloud covering the sky at a wavelength (W/m2nm)”. The description of “at a wavelength (W/m2nm)” is very confused to readers. The authors should be revised and give the information at what wavelength is.

Authors response:

I appreciate your suggestion. It was confusing, and we corrected as I_λ is the global spectral irradiance calculated by our spectrum model covering all-weather at a wavelength. The unit of I_λ (is W/m2nm)., The same correction was applied to other parameters so that the entire text is;

where, I_λ is the global spectral irradiance calculated by our spectrum model covering all-weather at a wavelength. The unit of I_λ is W/m2nm. f is the weather correction factor defined by Equation (2), I_1λ is the global spectral irradiance calculated using Bird’s spectrum model [98] at a wavelength. The unit of f is W/m2nm. I_2λ is the global spectral irradiance calculated by a spectrum model assuming full cloud covering the sky at a wavelength. The unit of I_2λ is W/m2nm. DNI is the direct normal irradiance. And, I_dλ is the direct normal solar spectral at a wavelength. The unit of I_dλ is W/m2nm.

Page 4 of 11, Fig. 3: the measured and estimated irradiances in the summertime are high than that of global irradiances (AM 1.5G) at the wavelength range of 450-550 nm. Please give the reasons that the irradiances are higher at this wavelength range.

Authors response:

I appreciate your suggestion. The reason was lower aerosol density in summer. An additional figure showing the seasonal trend of atmospheric parameters was added. The corrected text is;

Figure 4 indicates the measured aerosol density and precipitable water by the curve fitting to the measured solar spectrum. The aerosol optical depth was much lower than the standard value used to the calculation of the reference spectrum in summer (Figure 4 (a)). This is why the measured and estimated irradiances in the summertime are high than that of global irradiances (AM 1.5G) at the wavelength range of 450-550 nm. The precipitable water was much higher than the standard value used to the calculation of the reference spectrum in summer (Figure 4 (b)). This is why the measured and estimated irradiance dips are broader than that of global irradiances (AM 1.5G) at the wavelength range of 1350-1450 nm and around 1150 nm.

Figure 4. Seasonal trend of atmospheric parameters in Miyazaki [99].

 English was polished as well.

Round 2

Reviewer 1 Report

-